# Parallel Evolution of *Pseudomonas aeruginosa* during a Prolonged ICU-Infection Outbreak

David R. Cameron,[a] Melissa Pitton,[a,b] Simone Oberhaensli,[c] Katja Schlegel,[d] Guy Prod'hom,[e] Dominique S. Blanc,[f] Stephan M. Jakob,[a] Yok-Ai Que[a]

[a]Department of Intensive Care Medicine, Inselspital, Bern University Hospital, University of Bern, Bern, Switzerland
[b]Graduate School for Cellular and Biomedical Sciences (GCB), University of Bern, Bern, Switzerland
[c]Interfaculty Bioinformatics Unit and SIB Swiss Institute of Bioinformatics, University of Bern, Bern, Switzerland
[d]Institute of Psychology, University of Bern, Bern, Switzerland
[e]Institute of Microbiology, Lausanne University Hospital and University of Lausanne, Lausanne, Switzerland
[f]Service of Hospital Preventive Medicine, Lausanne University Hospital and University of Lausanne, Lausanne, Switzerland

**ABSTRACT** Most knowledge about *Pseudomonas aeruginosa* pathoadaptation is derived from studies on airway colonization in cystic fibrosis; little is known about adaptation in acute settings. *P. aeruginosa* frequently affects burned patients and the burn wound niche has distinct properties that likely influence pathoadaptation. This study aimed to genetically and phenotypically characterize *P. aeruginosa* isolates collected during an outbreak of infection in a burn intensive care unit (ICU). Sequencing reads from 58 isolates of ST1076 *P. aeruginosa* taken from 23 patients were independently mapped to a complete reference genome for the lineage (H25338); genetic differences were identified and were used to define the population structure. Comparative genomic analysis at single-nucleotide resolution identified pathoadaptive genes that evolved multiple, independent mutations. Three key phenotypic assays (growth performance, motility, carbapenem resistance) were performed to complement the genetic analysis for 47 unique isolates. Population structure for the ST1076 lineage revealed 11 evolutionary sublineages. Fifteen pathoadaptive genes evolved mutations in at least two sublineages. The most prominent functional classes affected were transcription/two-component regulatory systems, and chemotaxis/motility and attachment. The most frequently mutated gene was *oprD*, which codes for outer membrane porin involved in uptake of carbapenems. Reduced growth performance and motility were found to be adaptive phenotypic traits, as was high level of carbapenem resistance, which correlated with higher carbapenem consumption during the outbreak. Multiple prominent linages evolved each of the three traits in parallel providing evidence that they afford a fitness advantage for *P. aeruginosa* in the context of human burn infection.

**IMPORTANCE** *Pseudomonas aeruginosa* is a Gram-negative pathogen causing infections in acutely burned patients. The precise mechanisms required for the establishment of infection in the burn setting, and adaptive traits underpinning prolonged outbreaks are not known. We have assessed genotypic data from 58 independent *P. aeruginosa* isolates taken from a single lineage that was responsible for an outbreak of infection in a burn ICU that lasted for almost 2.5 years and affected 23 patients. We identified a core set of 15 genes that we predict to control pathoadaptive traits in the burn infection based on the frequency with which independent mutations evolved. We combined the genotypic data with phenotypic data (growth performance, motility, antibiotic resistance) and clinical data (antibiotic consumption) to identify adaptive phenotypes that emerged in parallel. High-level carbapenem resistance evolved rapidly, and frequently, in response to high clinical demand for this antibiotic class during the outbreak.

**KEYWORDS** adaptative evolution, antibiotic resistance, carbapenems, *oprD*, outer membrane porin D

Address correspondence to Yok-Ai Que, Yok-Ai.Que@insel.ch, or David R. Cameron, davcam32@gmail.com.

The authors declare no conflict of interest.

*P*seudomonas aeruginosa is a ubiquitous environmental bacterium that persists in a range of niches, including soil and water. *P. aeruginosa* is also one of the more versatile opportunistic human pathogens capable of causing a range of acute and chronic infections that often withstand antibiotic chemotherapy. *P. aeruginosa* adapts to the host environment, modulating the expression of numerous virulence factors and acquiring or developing means for antibiotic resistance. Understanding precisely how *P. aeruginosa* evolves during infection may lead to the identification of antibiotic or antivirulence targets that can be exploited for future therapy.

Pathoadaptive traits are those that are likely to improve bacterial fitness in a novel environment (1). Most of what is currently known about *P. aeruginosa* pathoadaptation (and bacterial pathoadaptation in general) comes from studies focused on chronic airway infection in the context of cystic fibrosis (CF) where individuals can be repeatedly sampled for decades (2–6). Environmental isolates colonize the CF airway and evolve to establish oftentimes incurable infections (7). Infections are associated with increased tolerance or resistance to the immune system and antibiotics (8–10), auxotrophy for specific amino acids that are abundant in patient sputum (11), formation of small colony variants (SCVs) (12), loss of motility (8), and the overproduction of alginate leading to mucoidy (13), which are postulated to evolve in response to CF-niche specific stressors that include inadequate antibiotic exposure, nutrient and oxygen availability, and the presence of other microorganisms (for review, see reference 14). In contrast, pathoadaptation of *P. aeruginosa* in acute infection settings is not well characterized.

*P. aeruginosa* is the most common Gram-negative pathogen isolated from infected burns and its presence is associated with significant mortality (15). The burn wound infection setting has unique properties that likely influence bacterial physiology and adaptive evolution. Burn wounds differ from nonburn-trauma wounds as their coagulation necrosis zones lack sufficient blood supplies and contain increased oxygen reactive species that impair wound healing (16, 17), which increases the risk for infection (18). In addition, burn wounds are characterized by a specific microenvironment composed of exudates that, together with necrotic materials, creates a niche for opportunistic pathogens such as *P. aeruginosa* (19). *P. aeruginosa* readily forms hard-to-treat biofilms within burn wounds (20), the formation of which often requires, among other factors, quorum sensing (21, 22), the acquisition of iron, and the excretion of exopolysaccharides, including alginate (23). The temporal evolution of pathogenic traits for *P. aeruginosa* affecting burn patients, however, is unknown.

In the current report, using genome sequencing data of 58 *P. aeruginosa* isolates from a single lineage that caused a well characterized outbreak of infections in a burn intensive care unit (24, 25), we used comparative genomics to identify pathoadaptive traits associated with successful adaptation to the burn wound infection milieu.

## RESULTS

**Outbreak of *Pseudomonas aeruginosa* infection in a burn intensive care unit.** We investigated a lineage of *P. aeruginosa* from sequence type ST1076 that was found previously to be the cause of an infection outbreak in the burn intensive care unit (ICU) at the Lausanne University Hospital (24). A total of 58 isolates from the lineage were collected from 23 patients between May 2010 and October 2012 (median of two isolates per patient, range 1 to 10) (Fig. 1A). One isolate (H25473) was excluded from the analysis due to a discrepancy between whole-genome sequencing data and the actual genotype/phenotype of the bacterial isolate (see Table S1 for more details). To determine the population structure of the lineage, sequencing reads from each isolate ($n = 57$) were mapped to the complete genome sequence of the earliest isolate, H25883, and comparative genomics was performed. The mean number of reads mapped ($6.4 \times 10^6$) and the mean sequencing depth ($112 \times$) were high (Fig. S3A), as were the mean coverage of the complete H25883 genome (99.95%) and the mean pairwise identity for the mapped reads (98.71%; Fig. S3B), confirming the veracity of read mapping to a single reference for subsequent comparative genomic approaches. Two complementary methods (snippy and Geneious Prime) were used to identify 113 unique genomic differences across the collection

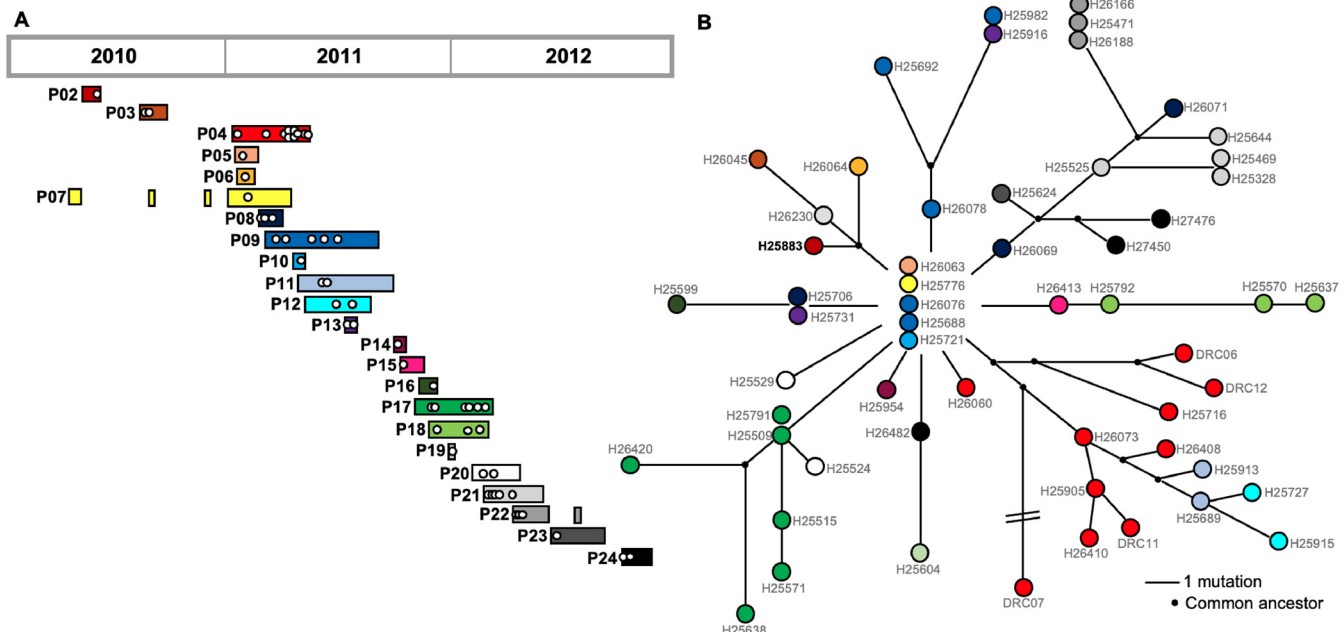

**FIG 1** Population structure of *P. aeruginosa* isolates from an ICU outbreak of infection. (A) *P. aeruginosa* isolates from a lineage of ST1076 were collected from 23 infected patients during the outbreak in the burn ICU (2010 to 2012). Patient 01 (P01) was not included as the ST1076 isolate was only distantly related to those from the outbreak. Each white circle represents a single isolate. Bars indicate patient time in the ICU (adapted from reference 25). (B) Population structure was inferred from whole-genome sequencing data (Table S2) to create a minimum spanning tree. Sequence reads were mapped to the complete genome of H25883 (in bold font). A common ancestor was inferred when subbranches had a shared mutation. The color of each circle corresponds to the patient shown in panel A.

(Table S2); 12 were insertions, 29 were deletions, and 72 were single nucleotide polymorphisms (SNPs). Of these, 66 were nonsynonymous and six were synonymous. Of the deletions, six were greater than 100 basepairs (bp) in size (largest deletion, 21,140bp). In total, 108 predicted genes (coding DNA sequences, CDS) were mutated at least once (1.7% of all CDS).

There were 6.9 genomic differences per isolate on average (range 2 to 17; Table S3). A mutation matrix (Table S2) was used to construct a population structure that revealed a core consisting of five identical isolates from four patients which likely represents the ancestral clone (Fig. 1B). From there, 11 distinct branches were identified which will be hereby referred to as sublineages (median 4 isolates per sublineage, range 1 to 13; Fig. 1B). Interestingly, some isolates were collected before the "core" clone was identified. This could be explained by the persistence of ST1076 on various surfaces within the burn ICU (24). The core clone was introduced in multiple patients via this source; isolates collected before identification of the core clone had likely already begun to acquire niche specific adaptations in the patients.

**Parallel evolution of 15 "pathoadaptive" genes.** Pathoadaptive genes were defined as those that were mutated with nonsynonymous changes at least twice independently, as parallelism is a useful signal of adaptive evolution (26). Fifteen genes were classified as pathoadaptive (Fig. 2). Using PseudoCAP classifications (27), most of the genes were classed as transcriptional regulators or two-component regulatory systems (*n* = 8, 53%). The second most frequent predicted function for the pathoadaptive genes was involvement in motility and attachment (*n* = 3, 20%). Functional classes "antibiotic susceptibility," "adaptation, protection," "cell wall/LPS/capsule," and "transport of small molecules" each included two pathoadaptive genes. The most frequently mutated gene was *oprD*, which codes for outer membrane porin D.

**Reduced growth rate and motility for isolates adapted to the burn infection setting.** Adaptation of *P. aeruginosa* to different niches is often accompanied by changes in growth performance. Adaptation to standard laboratory conditions is associated with enhanced *in vitro* fitness and increased growth rate (28, 29). Conversely, long-term adaptation in chronic infection settings such as CF is associated with reduced growth rate (29). To determine the effects of adaptation upon *in vitro* fitness in the context of acute infection, we monitored the $OD_{600}$ of 47 unique isolates in liquid culture across time. A range of temporal growth

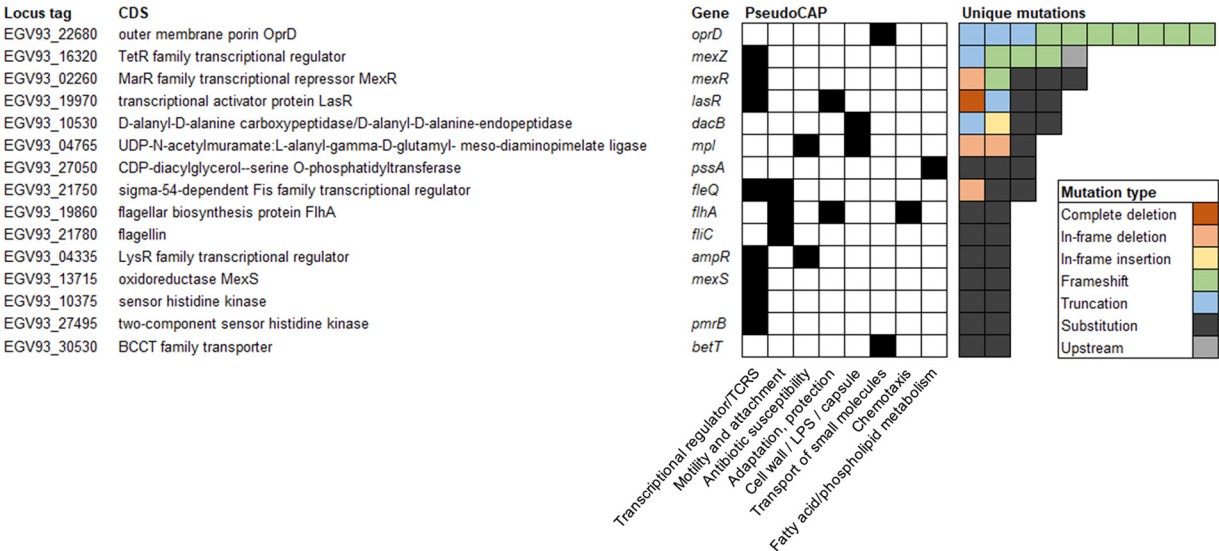

| Locus tag | CDS | Gene |
|---|---|---|
| EGV93_22680 | outer membrane porin OprD | *oprD* |
| EGV93_16320 | TetR family transcriptional regulator | *mexZ* |
| EGV93_02260 | MarR family transcriptional repressor MexR | *mexR* |
| EGV93_19970 | transcriptional activator protein LasR | *lasR* |
| EGV93_10530 | D-alanyl-D-alanine carboxypeptidase/D-alanyl-D-alanine-endopeptidase | *dacB* |
| EGV93_04765 | UDP-N-acetylmuramate:L-alanyl-gamma-D-glutamyl- meso-diaminopimelate ligase | *mpl* |
| EGV93_27050 | CDP-diacylglycerol--serine O-phosphatidyltransferase | *pssA* |
| EGV93_21750 | sigma-54-dependent Fis family transcriptional regulator | *fleQ* |
| EGV93_19860 | flagellar biosynthesis protein FlhA | *flhA* |
| EGV93_21780 | flagellin | *fliC* |
| EGV93_04335 | LysR family transcriptional regulator | *ampR* |
| EGV93_13715 | oxidoreductase MexS | *mexS* |
| EGV93_10375 | sensor histidine kinase | *pmrB* |
| EGV93_27495 | two-component sensor histidine kinase | *pmrB* |
| EGV93_30530 | BCCT family transporter | *betT* |

**FIG 2** Fifteen frequently mutated "pathoadaptive genes." Pathoadaptive genes were defined as those with at least two independent mutations across the isolate collection. CDS, coding DNA sequence; LPS, lipopolysaccharide; TCRS, two-component regulatory system.

dynamics were observed (Fig. 3A), which translated into a bimodal distribution for our measure of growth rate, $\Delta OD_{600}$ per hour (Fig. 3B). Isolates close to the core of the population structure (i.e., H26076) were typically fast growers, and slow growers emerged in most sublineages (six of 11; Fig. 3C), which suggests that growth rate decreases following adaptation to the burn infection setting.

We extended the functional class analysis performed for the pathoadaptive genes (Fig. 2) to include each of the 108 mutated genes across the collection (Table S4) and found that genes involved in "chemotaxis" and "motility and attachment" were overrepresented; 11% and 10% of all genes within each class, respectively, were mutated within the isolate collection (Fig. 4A). To functionally assess the impact of this mutational trend, we assessed swimming motility by stab inoculating each of the 47 unique isolates into 0.3% LB agar plates. As was observed for growth performance, isolates at the core of the population structure were motile, and isolates with impaired motility emerged in most sublineages (six of 11; Fig. 4B). In fact, there was a positive correlation between $\Delta OD_{600}$ per hour and swimming motility (motility was inferred by quantifying colony size in pixels$^2$, Pearson $r = 0.70$, $P < 0.0001$; Fig. S1).

**Carbapenem use in the ICU is associated with *oprD* mutation and carbapenem resistance.** Not only was *oprD* the most frequently mutated gene across the collection (Fig. 2A), the mutations detected were also highly impactful (seven frameshifts and three truncations; Fig. 5A). OprD is responsible for influx of carbapenem antibiotics, including meropenem and imipenem (30). Further, *oprD* mutation is a primary resistance mechanism when *P. aeruginosa* is exposed to carbapenems *in vitro* (31). We hypothesized that *oprD* mutation during the *P. aeruginosa* ICU outbreak occurred in response to *in vivo* carbapenem exposure and that isolates with mutated *oprD* would have elevated MIC toward carbapenem class antibiotics. To test this, we first analyzed published data that reported antibiotic consumption in the same ICU burn center at the time of the *P. aeruginosa* infection outbreak (32). Leading up to the outbreak and throughout the sampling period, carbapenems were the most frequently administered class of antibiotic in the burn center (Fig. 5B). From 2004 to 2012, carbapenem use, reported as defined daily doses per 1,000 burn days, was higher than the use of other antipseudomonal antibiotic classes, including aminoglycosides ($P < 0.0001$), quinolones ($P < 0.0001$), cephalosporins ($P < 0.0001$), colistin ($P < 0.0001$), and penicillins (namely, piperacillin-tazobactam, $P = 0.003$, each $P$-value was determined using two-way ANOVA with Tukey's multiple-comparison test).

Meropenem and imipenem MICs were determined for each unique *P. aeruginosa* isolate. Using EUCAST susceptibility breakpoints, 24 of the isolates were susceptible to meropenem

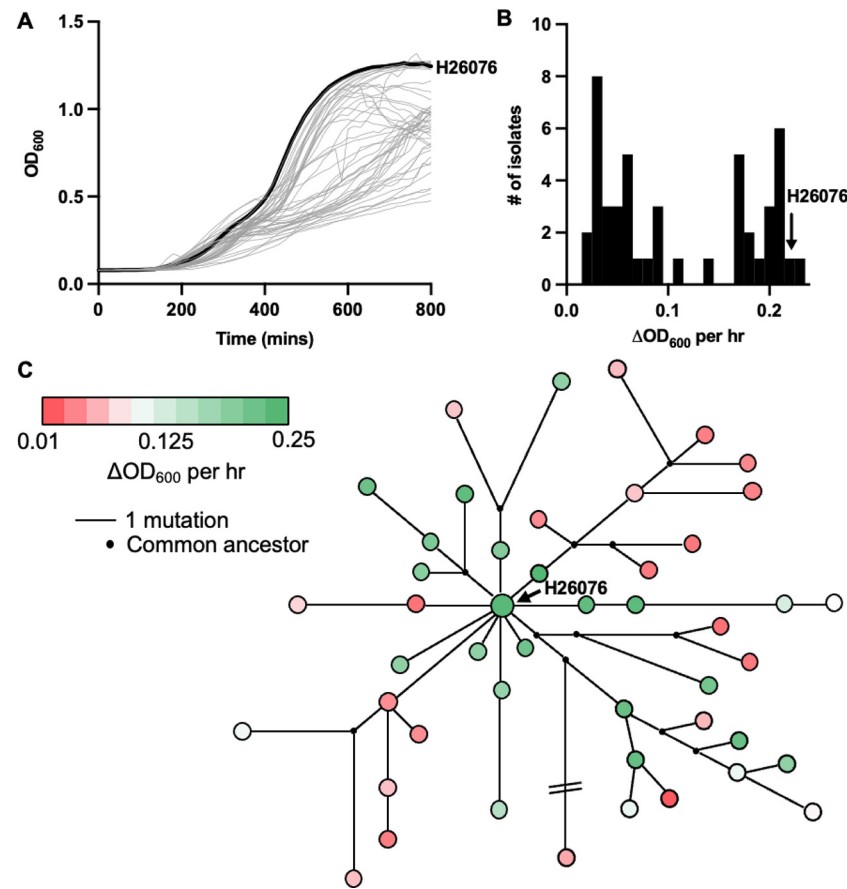

**FIG 3** Reduced growth rate is an adaptive trait for *P. aeruginosa* lineages evolving during the outbreak in the burn ICU. (A) Temporal *in vitro* growth dynamics of *P. aeruginosa* from the burn ICU. An isolate at the core of the population structure, H26076, is presented in bold text. (B) Bimodal distribution of isolates based on the change in optical density at 600 nm ($OD_{600}$) per hour during the exponential phase of growth. (C) Change in $OD_{600}$ per hour in the context of population structure for the *P. aeruginosa* lineage.

(MIC ≤2 µg/mL), three were susceptible at increased exposure (MIC between 2 and 8 µg/mL) and 20 were resistant (MIC > 8 µg/mL) (Fig. 5C; Table S1). In contrast, none of the isolates were fully susceptible to imipenem (MIC ≤0.001 µg/mL), 22 were susceptible at increased exposure (MIC >0.001, ≤4 µg/mL) and 25 were resistant (MIC >4 µg/mL; Table S1). MIC values for meropenem and imipenem were positively correlated ($r = 0.7070$, $P < 0.0001$, Pearson correlation), and as such, only meropenem MICs will be used/discussed further. Each isolate with high meropenem MIC (i.e., susceptible at increased exposure or resistant, >2 µg/mL) had an *oprD* mutation and the isolates were spread across six of the 11 distinct sublineages (55%; Fig. 5D).

The list of pathoadaptive mutations included additional genes known to confer antibiotic resistance when mutated, including *mexZ*, *mexR*, *mexS*, *dacB*, *ampR*, *pmrB* (33–37) (Fig. 2). Accordingly, resistance to additional antibiotic classes, including penicillins, cephalosporins, and quinolones was observed across the collection, albeit less frequently than for carbapenems (Fig. S4).

**Parallel genetic mutation correlated with phenotypic convergence.** To test whether parallel evolution of the 11 sublineages (Fig. 6A) correlated with phenotypic convergence toward adaptive traits (slow growth, loss of motility, elevated meropenem MIC), we performed a cluster analysis and visualized the data in three dimensions (Fig. 6B). Each of the phenotypically characterized isolates ($n = 47$) was categorized within four robust clusters (good cohesion and separation; Fig. S2) that were predicted *a priori*: cluster A included 12 isolates with comparatively high growth rate (mean $\Delta OD_{600}$/h 0.20 +/− standard deviation [SD] 0.02) and swimming motility (0.61 pixels² +/− 0.18), and low meropenem MIC

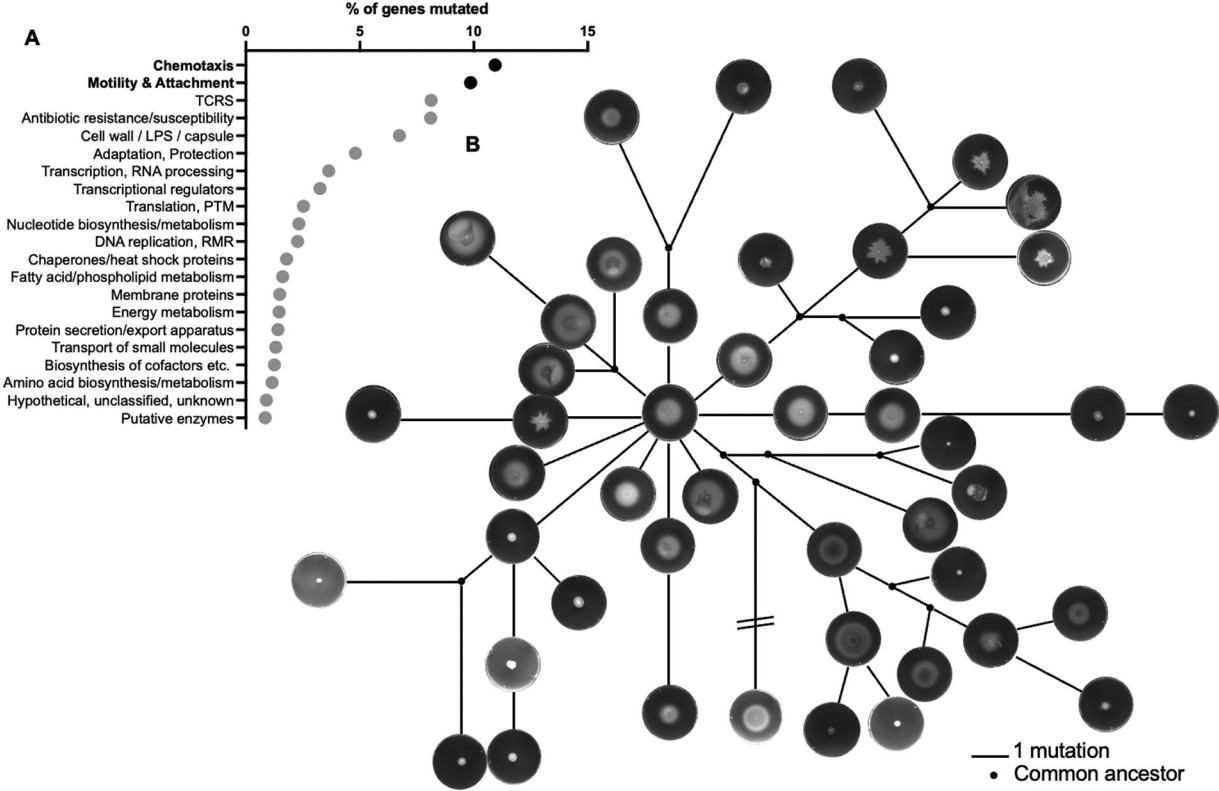

**FIG 4** Motility is an adaptive trait for *P. aeruginosa* in the burn infection setting. (A) Each of the 108 mutated genes was functionally classified using PseudoCAP. A percentage of genes' mutated measure was determined by relating the number of mutated genes to the total number of genes within each functional category. (B) Each unique *P. aeruginosa* isolate was stab inoculated into 0.3% LB agar plates. Plates were imaged after 24 h of incubation at 37°C. Images of each plate was then plotted in the context of population structure. PTM, posttranslational modification; LPS, lipopolysaccharide; RMR, recombination, modification, repair; TCRS, two-component regulatory system.

(median 0.19 $\mu$g/mL, 95% confidence interval [CI] = 0.125 to 0.38); cluster B included 11 isolates with low growth rate ($\Delta OD_{600}$/h 0.46 $\pm$ 0.02) and swimming motility (0.09 pixels$^2$ $+/-$ 0.09) and low meropenem MIC (0.25 $\mu$g/mL 95%CI = 0.19 to 0.75); cluster C included nine isolates with comparatively high growth rate ($\Delta OD_{600}$/h 0.17 $\pm$ 0.06) and swimming motility (0.51 pixels$^2$ $+/-$ 0.28), and high meropenem MIC ($>$32 $\mu$g/mL 95% CI = 8 to $>$32 [the highest concentration for meropenem Etest is 32 $\mu$g/mL]); cluster D included 15 isolates with low growth rate ($\Delta OD_{600}$/h 0.05 $\pm$ 0.03) and swimming motility (0.13 pixels$^2$ $+/-$ 0.14), and high meropenem MIC ($>$32 $\mu$g/mL, 95% CI = $>$32 to $>$32) (Fig. S5; Table S1). Adapted clusters (B, C, D) each had representatives from multiple sublineages (three, five, and five sublineages, respectively) suggesting that the adaptive traits were evolving in parallel (Fig. 6C).

**Other adaptive traits in burn infection.** Of note, one of the more frequently mutated pathoadaptive genes was *lasR* (four independent mutations; Fig. 2) which correlated with enhanced blue-pigment and colony autolysis on agar plates (Fig. S6A). Additionally, while not classed as pathoadaptive (i.e., only one independent mutation across the collection), two isolates harbored *mucA* mutation which corresponded with a mucoidy phenotype on agar plates (Fig. S6B).

## DISCUSSION

*P. aeruginosa* is a notorious cause of acute infections in hospital settings, particularly those afflicting burn patients. The infection outbreak that serves as the focus of this study is a perfect example; epidemiological studies identified a dominant lineage belonging to ST1076 that persisted in the hospital environment (specifically on sink traps and mattresses in the hydrotherapy room) and spread between 23 patients in a burn ICU (24, 25). As a complement to these findings, we focused our analysis on specific gene mutations

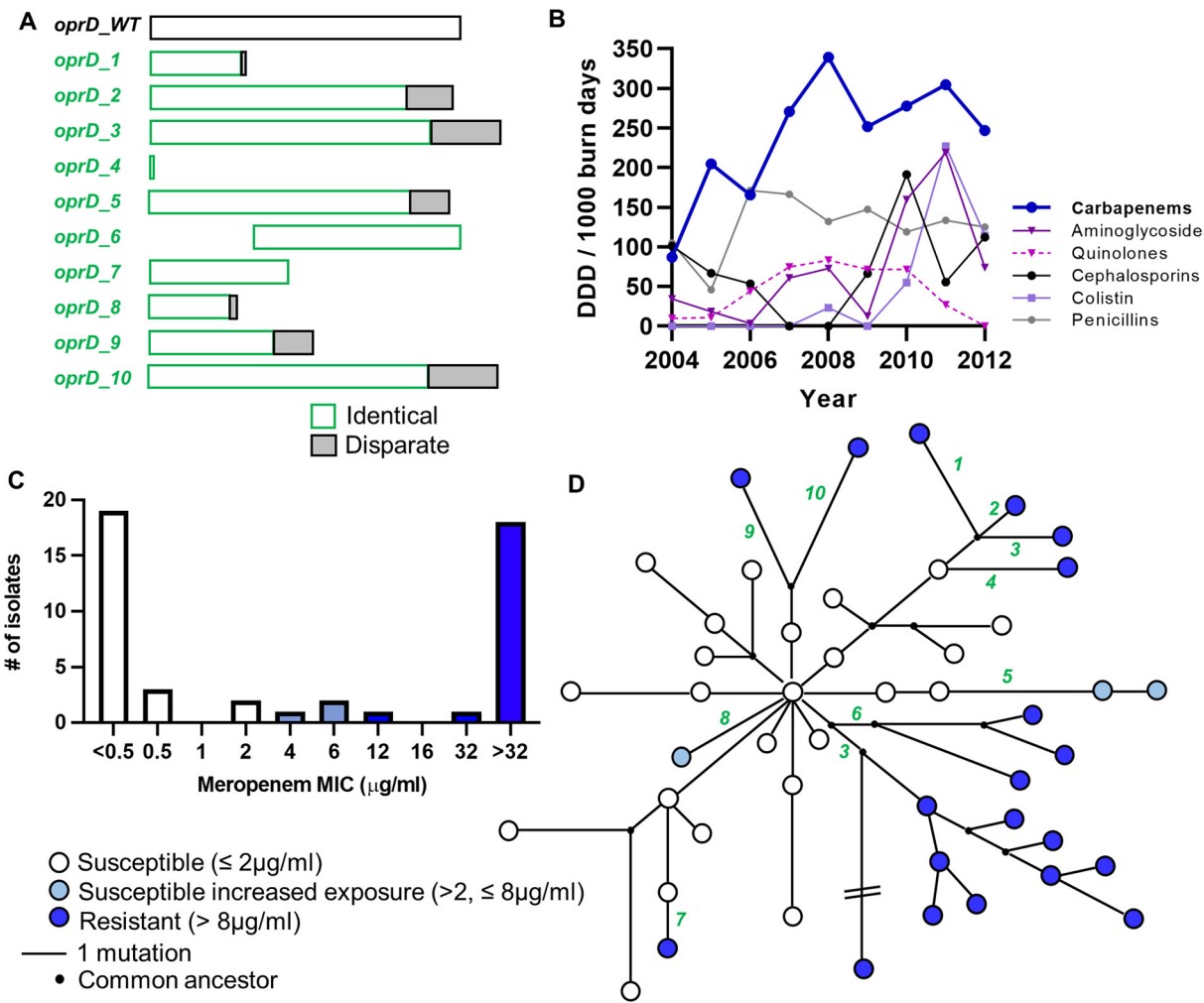

**FIG 5** OprD mutation is associated with carbapenem use in the ICU and the emergence of carbapenem resistance. (A) Schematic representation of the impact of *oprD* mutation to predicted OprD amino acid sequence. *oprD_WT* is the "wild-type" unmutated gene (translates to a sequence 444 amino acids in length) and *oprD_1* through to *oprD_10* are mutations that emerged during the infection outbreak. Regions in green show 100% identity with *oprD_WT* sequence. Regions in gray are different to *oprD_WT* (disparate region). (B) Defined daily dose (DDD) of antibiotics from different classes used before and during the ICU infection outbreak (adapted from reference 31). (C) Distribution of meropenem MICs across tested isolates (*n* = 47) as determined by Etest (the highest concentration in the meropenem Etest is 32 µg/mL). The color corresponds to clinical breakpoints defined by the European Committee on Antimicrobial Susceptibility Testing (EUCAST). (D) Meropenem MIC plotted in the context of population structure. Each green number corresponds to the OprD schematic from panel A.

and pathoadaptive traits that may have underscored the success of the lineage. We found evidence for parallel evolution across distinct sublineages, most notably toward reduced growth performance, loss of a virulence determinant (motility), and antibiotic resistance.

Reduced growth rates are a hallmark of well-adapted *P. aeruginosa* strains, particularly in the context of CF where growth *in situ* has been measured (38). Slow growing bacteria can emerge due to a wide range of mutations, and they are difficult to kill with conventional antibiotics, each of which may explain their frequent selection *in vivo* (39–41). It has been proposed that antibiotic tolerance facilitated by slow growth provides an adequate opportunity to support the development of targeted mutations that are responsible for high level antibiotic resistance (29). Our study provides some support for this notion; five of the 10 independent introductions of *oprD* mutation and elevated carbapenem MICs were preceded by a slow growing isolate, and slow growth and antibiotic resistance emerged concurrently for two more. For the remaining three, however, *oprD* mutation was not preceded by or associated with slow growth.

We have shown that in a "real-world" outbreak scenario, motility is a dispensable trait, and that its loss may provide a selective advantage during persistent infection. This appears

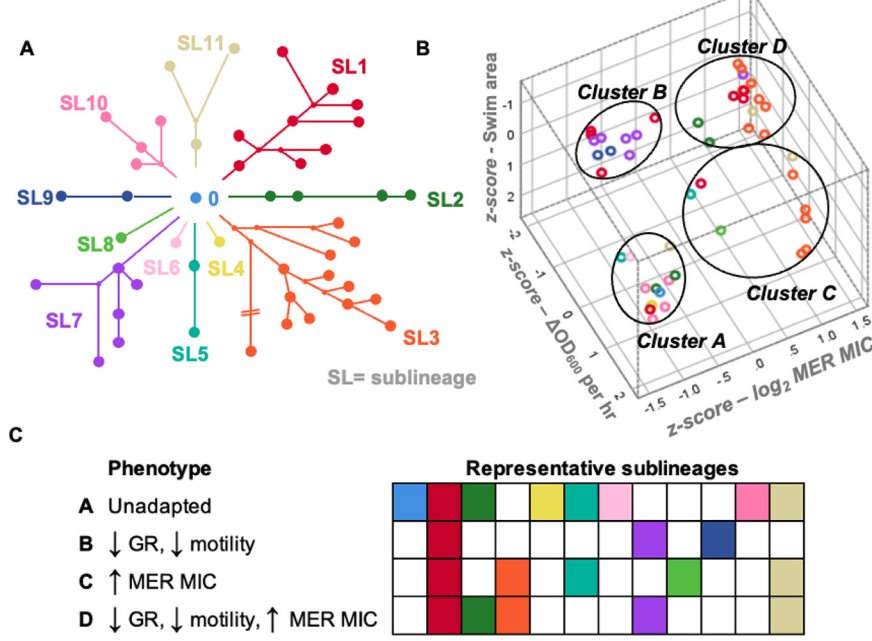

**FIG 6** Parallel evolution of adaptive traits for *P. aeruginosa* sublineages isolated during an ICU infection outbreak. (A) Eleven distinct sublineages were identified based on population structure. (B) TwoStep cluster analysis was used to identify four distinct phenotypic clusters (clusters A to D). (C) Each phenotypic cluster contained representative isolates from a range of sublineages. Conversely, most sublineages contained isolates from multiple phenotypic clusters. GR, growth rate (GR), as inferred from $\Delta OD_{600}$ per hour; MER MIC, meropenem MIC.

in contrast with findings from multiple reports that have highlighted the importance of motility for full pathogenesis in models of burn infection due to *P. aeruginosa* (42–44). Taken together, a plausible explanation may be that motility is required only during the early stages of pathoadaptation, and that its loss could be beneficial over the course of infection (or infection outbreak) by limiting, for example, phagocytosis (45).

Perhaps the most significant finding of the comparative genomic analysis was the rapid and repeated evolution toward carbapenem resistance facilitated by impactful *oprD* mutations. The mutational response correlated with the high clinical use of antibiotics from the carbapenem class. These findings are in accordance with a separate molecular study that identified loss of OprD as the primary driver of antibiotic resistance emergence for *P. aeruginosa* isolated from ICU patients at a separate center (46). Similarly, a recent study highlighted the rapid emergence of *oprD* mutants after administration of meropenem in a patient with acute lung infection; however, these mutants were eventually replaced by isolates with mutated MexAB-OprM (47).

Although the sampling period for the current study was short (∼2.5 years), a remarkable number of pathoadaptive traits were identified; in addition to reduced growth rate, loss of motility, and antibiotic resistance, we also identified *lasR* mutants with phenotypes consistent with altered quorum sensing, as well as mucoid variants (Fig. S6), all of which have been described previously in the context of chronic CF infection (13, 48). These findings, taken with a recent report showing SCV emergence in an infected burn patient (49), suggest that while the niches are fundamentally distinct, evolutionary pathways toward infection are similar between acute (i.e., burn) and chronic (i.e., CF) settings. Additionally, while sampling in CF studies has been performed over many decades, it is apparent that significant adaptation occurs in the first 2 to 3 years (50), which is similar to the time frame assessed in this study, and further highlights the rapid nature of *P. aeruginosa* reprogramming for persistence in the human host.

In summary, we have characterized the pathoadaptation of a single lineage of *P. aeruginosa* taken from human patients hospitalized in a burn ICU during an outbreak of infection.

Future studies are certainly warranted to determine if the same adaptive traits identified here are common and convergent across distinct lineages in acute and/or outbreak infection settings.

## MATERIALS AND METHODS

**Bacterial isolates and growth conditions.** *P. aeruginosa* isolates analyzed in this report ($n = 58$) were collected from hospitalized patients between 2010 and 2012 in the context of previous epidemiological studies (24, 25) of an outbreak of infection in a burn ICU (Table S1). Authorization for analyzing anonymized bacterial isolates and anonymized already published data were not required. Phenotypic experiments were performed for 47 isolates with unique genotypes. *P. aeruginosa* were routinely propagated in Luria-Bertani (LB; BD, NJ, USA) broth at 37°C with constant shaking.

**Comparative genomics.** For 54 isolates, illumina sequencing reads generated in a previous study (25) were downloaded from the Sequence Read Archive (SRA) public database using the SRA accession numbers listed in Table S1. For an additional four isolates (see Table S1), complete genomes were generated as follows: genomic DNA was extracted using a DNeasy Ultraclean Microbial kit (Qiagen, Hilden, Germany) then sequenced using PacBio RS and Illumina NovaSeq 6000 sequencing platforms. PacBio long reads were used to assemble a single, circular chromosome using Flye version 2.6 (51) and the assembly was polished with Illumina short read data using Pilon (52). Complete genome sequences were annotated using the NCBI Prokaryotic Genome Annotation Pipeline (53) and were submitted to DDBJ/ENA/GenBank, and illumina reads were submitted to SRA, each using accession numbers listed in Table S1.

Two complementary strategies were used to identify mutations at single nucleotide resolution, each involving the use of the high-quality complete reference strain H25338 (54). First, SNPs and small insertion deletion (InDel) mutations were detected for each isolate using snippy (https://github.com/tseemann/snippy). Second, to validate the snippy results and to search for large deletions, reads were mapped to the reference using Geneious Prime software (version 2022.1.1). Genomic differences were detected using the Find Variations/SNPs function (minimum coverage 10, minimum variant frequency 0.75, maximum variant *P*-value $10^{-6}$), and mapping was assessed visually. A mutation matrix was generated (Table S2) which was used to infer phylogeny and generate a minimum spanning tree using PHYLOViZ (55), which implements the goeBURST algorithm (56).

**Growth assays.** Overnight cultures of *P. aeruginosa* were diluted 1:1,000 in 150 $\mu$L volumes of sterile LB media in 96-well microtiter plate wells. Plates were incubated at 37°C and optical density (OD) at 600 nm was measured every 15 min using a SpectraMax i3 system (Molecular Devices, CA, USA). Plates were shaken prior to each measurement. Assays were performed in biological duplicate and the mean was used to generate growth curves. To provide a measure of growth rate, a 3-h period within the exponential phase of growth (between 5.5 h and 8.5 h) was used to infer the change in OD per hour using the following equation: $\Delta OD_{600}/hr = OD_{600}$ at 8.5 h $- OD_{600}$ at 5.5 h/3.

**MIC determinations.** Clinical antibiotic susceptibility data were generated at the Institute of Microbiology of the University of Lausanne. MICs for meropenem and imipenem were determined on Mueller-Hinton agar using Etest (bioMérieux, France) according to the manufacturer's instruction. MIC values were recorded following 24 h of incubation at 37°C. MIC breakpoints were defined by the European Committee on Antimicrobial Susceptibility Testing (EUCAST v12.0).

**Motility assays.** Swimming motility was assessed using previously described methods (57). Briefly, a sterile toothpick was dipped into overnight cultures of *P. aeruginosa* and then used to stab inoculate the center of 0.3% LB agar plates. Plates were incubated at 37°C for 24 h and then imaged using an iBright 750 imaging system (Invitrogen, USA). Quantification of swimming area was performed using imageJ (58).

***oprD* sequencing.** The *oprD* was amplified by PCR using primers (5′ to 3′) GAACCTCAACTATCGCCAAG and TGTCGGTCGATTACAGGATC. The *oprD* containing fragment was sequenced via sanger sequencing using primers (5′ to 3′) CAAGAAGAACTAGCCGTCAC, GCTACGCAATCACCGATAAC, and GGATCGACAGCGGATAGTC.

**Cluster analysis.** Growth, motility, and $\log_2$ transformed meropenem MIC data (Table S1) were used to classify each isolate into phenotypic clusters. First, correlations between one variable (i.e., MIC) and a second variable (i.e., growth) were determined (Fig. S1). A significant positive correlation existed between growth rate and motility, leading to the *a priori* hypothesis that phenotypic data are likely to result in four clusters: (i) unadapted; (ii) low growth, low motility; (iii) high meropenem MIC; (iv) low growth, low motility, high meropenem MIC. To test this, data were z-transformed and clusters were identified using the two-step cluster algorithm specifying detection of four clusters. Cluster quality received the highest classification ("good," score $> 0.5$) based on the Silhouette measure of cohesion and separation (Fig. S2). Analyses were performed in SPSS (IBM, version 25).

## SUPPLEMENTAL MATERIAL

Supplemental material is available online only.

**SUPPLEMENTAL FILE 1**, PDF file, 0.6 MB.
**SUPPLEMENTAL FILE 2**, XLSX file, 0.02 MB.
**SUPPLEMENTAL FILE 3**, XLSX file, 0.04 MB.
**SUPPLEMENTAL FILE 4**, XLSX file, 0.1 MB.
**SUPPLEMENTAL FILE 5**, XLSX file, 0.02 MB.

## ACKNOWLEDGMENTS

The authors acknowledge the excellent technical assistance of Sandra Nansoz, Severin Jung, and Viola Grünenfelder. DNA whole-genome sequencing was performed at the Next Generation Sequencing Platform of the University of Bern. Sanger sequencing was performed at Microsynth.

The study was funded by a research grant from the Stiftung für die Forschung in Anästhesiologie und Intensivmedizin (Nr. 32/2019 awarded to D.R.C.). The funders had no role in study design, data collection and analysis, decision to publish, or preparation of the manuscript.

The authors have declared that no competing interests exist.

D.R.C. and Y.-A.Q. designed the study. D.R.C. analyzed the genetic data. S.O. provided bioinformatic support. D.R.C. and M.P. performed phenotypic experiments. K.S. performed statistical analyses. D.S.B. and G.P. provided the bacterial strains, antibiotic resistance, and epidemiological information about the infection outbreak. S.M.J. supervised the study. D.R.C. wrote the manuscript. All authors reviewed and edited the manuscript.

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
