## [Reviewer comments · Microbiology Spectrum]

Microbiology Spectrum

Parallel evolution of *Pseudomonas aeruginosa* during a prolonged ICU infection outbreak

David Cameron, Melissa Pitton, Simone Oberhaensli, Katia Schlegel, Guy Prod'hom, Dominique Blanc, Stephan Jakob, and Yok-Ai Que

Corresponding Author(s): Yok-Ai Que, Inselspital

Review Timeline:

Submission Date:	July 18, 2022
Editorial Decision:	August 30, 2022
Revision Received:	September 16, 2022
Accepted:	October 19, 2022

Editor: Joanna Goldberg

Reviewer(s): The reviewers have opted to remain anonymous.

Transaction Report:

DOI: <https://doi.org/10.1128/spectrum.02743-22>

August 30, 2022

Prof. Yok-Ai Que
Inselspital
Bern
Switzerland

Re: Spectrum02743-22 (Parallel evolution of *Pseudomonas aeruginosa* during a prolonged ICU infection outbreak)

Dear Prof. Yok-Ai Que:

Thank you for submitting your manuscript to Microbiology Spectrum. Your paper was reviewed by two experts that both found the work of interest. However each had comments to help improve the paper that should be considered in a resubmission. These are listed below.

Link Not Available

Sincerely,

Joanna Goldberg

Journals Department
Reviewer comments:

Reviewer #1 (Comments for the Author):

The manuscript describes detailed genomic and phenotypic analyses of 58 ST1076 *Pseudomonas aeruginosa* isolates from an outbreak in the burn unit of a single medical center. A total of 16 genes were found to have acquired mutations independently (i.e., in at least two lineages from the outbreak), suggesting that they were pathoadaptive (conferred a fitness advantage). The most frequently mutated gene was *oprD*, and these mutations were shown to occur in the context of frequent carbapenem use in the ICU. Reduced growth rate and reduced motility were also frequently observed. The manuscript is clearly and concisely written and presents interesting findings on how *P. aeruginosa* evolves over the course of an outbreak. A few clarifications and modifications would increase its significance.

1. Materials and methods (lines 131-132). More information should be provided on how the phylogenetic trees were generated. The software used to generate the trees is listed, but information should also be given about the type of trees that were made (distance method?).
2. Materials and methods (lines 150-153). It is stated that oprD mutations were determined by PCR followed by Sanger sequencing. Why were the whole-genome sequences not used for this purpose?
3. Results: lines 184-188. Nonsynonymous vs synonymous mutations are provided in Table S2, but a discussion of how many SNPs were synonymous and how many were nonsynonymous should be provided in the Results section. Only nonsynonymous SNPs should be included in the subsequent discussions of genes that were mutated and potentially pathoadaptive, as synonymous SNPs would not be expected to have an effect on gene products.
4. Fig. 1: Some "core" isolates, from which all the other isolates were hypothesized to evolve from, were cultured several months after several of their supposed descendants. Some discussion of how this may have occurred should be provided.
5. Lines 194-195. Likewise, it should be clarified whether "pathoadaptive" genes were defined as those that were mutated at least twice (as stated in the text) or were mutated at least twice with nonsynonymous changes, which would be required if the genes are truly pathoadaptive.
6. Fig. S3B. An explanation for the two pairs of isolates with relatively low pairwise identity should be provided. If all these isolates shared a single ancestor, why are these two pairs relatively dissimilar?
7. Fig. 5. In panel A, this reviewer is not familiar with the terms "sense" and "nonsense" being used in this way. In panel B, different colors should be used for the different lines to make it easier to distinguish them.

Reviewer #2 (Comments for the Author):

Interesting work analyzing the evolution of *Pseudomonas aeruginosa* during a prolonged outbreak in burn ICU. There are however some comments for the authors consideration:

1. Lines 140-144. Why were only the MICs of imipenem and meropenem determined? Would be informative to add other relevant antipseudomonal agents.
2. Related to this, other relevant mutations related to efflux pumps expression (mexR, mexZ, mexS) or AmpC expression (dacB, mpl, ampR) are seen and should be interpreted since many of the mutated genes are related to the *P. aeruginosa* mutational resistome (see for instance López-Causapé *C front Microbiol* 2018).
3. Lines 240-245. The interpretation of EUCAST breakpoints and definitions is incorrect, particularly "intermediately susceptible" shall not be used since it is misleading.
4. One relevant phenotype that could/should be studied in the context of adaptation in burn wound infections is the capacity to form biofilms.
5. It might be informative to discuss your results together with those of Wheatley *R Nat Commun* 2021

Staff Comments:

Preparing Revision Guidelines

Please return the manuscript within 60 days; if you cannot complete the modification within this time period, please contact me. If you do not wish to modify the manuscript and prefer to submit it to another journal, please notify me of your decision immediately so that the manuscript may be formally withdrawn from consideration by Microbiology Spectrum.

Prof. Joanna Goldberg
Editor
Microbiology Spectrum

Bern, September 13th, 2022

Re: Response to review: Parallel evolution of *Pseudomonas aeruginosa* during a prolonged ICU infection outbreak

Dear Prof. Joanna Goldberg,

We would like to thank the reviewers for their positive comments and helpful suggestions. We have performed additional experiments and included more clinical data as requested. Below, we have included a point-by-point response to each comment.

Reviewer #1

The manuscript describes detailed genomic and phenotypic analyses of 58 ST1076 *Pseudomonas aeruginosa* isolates from an outbreak in the burn unit of a single medical center. A total of 16 genes were found to have acquired mutations independently (i.e., in at least two lineages from the outbreak), suggesting that they were pathoadaptive (conferred a fitness advantage). The most frequently mutated gene was *oprD*, and these mutations were shown to occur in the context of frequent carbapenem use in the ICU. Reduced growth rate and reduced motility were also frequently observed. The manuscript is clearly and concisely written and presents interesting findings on how *P. aeruginosa* evolves over the course of an outbreak. A few clarifications and modifications would increase its significance.

We appreciate the positive response of reviewer #1.

1. Materials and methods (lines 131-132). More information should be provided on how the phylogenetic trees were generated. The software used to generate the trees is listed, but information should also be given about the type of trees that were made (distance method?).

More information about the type of tree generated has been added to the Material and methods section (lines 126-128).

2. Materials and methods (lines 150-153). It is stated that *oprD* mutations were determined by PCR followed by Sanger sequencing. Why were the whole-genome sequences not used for this purpose?

Mutations in *oprD* were originally called using whole-genome sequencing data and then validated by PCR and Sanger sequencing.

3. Results: lines 184-188. Nonsynonymous vs synonymous mutations are provided in Table S2, but a discussion of how many SNPs were synonymous and how many were nonsynonymous should be provided in the Results section. Only nonsynonymous SNPs should be included in the subsequent discussions of genes that were mutated and potentially pathoadaptive, as synonymous SNPs would not be expected to have an effect on gene products.

We agree with the reviewer. The number of SNPs that were synonymous and nonsynonymous has been added to the Results section accordingly (lines 180-183).

4. Fig. 1: Some "core" isolates, from which all the other isolates were hypothesized to evolve from, were cultured several months after several of their supposed descendants. Some discussion of how this may have occurred should be provided.

The bacterial isolates were persisting on surfaces in the ICU throughout the outbreak. It is possible that by the time each respective isolate was collected, independent, niche specific evolution had already occurred. Some discussion has been added to the text (lines: 190-194).

5. Lines 194-195. Likewise, it should be clarified whether "pathoadaptive" genes were defined as those that were mutated at least twice (as stated in the text) or were mutated at least twice with nonsynonymous changes, which would be required if the genes are truly pathoadaptive.

We agree with this point and we have modified the definition of pathoadaptive genes as suggested by the reviewer (ie., mutated with nonsynonymous changes at least twice independently).

One of the 16 genes classified originally as pathoadaptive (*rhIR*) did not fulfil the criteria of the new definition. Therefore, both the text and the **Fig. 2** have been modified accordingly.

6. Fig. S3B. An explanation for the two pairs of isolates with relatively low pairwise identity should be provided. If all these isolates shared a single ancestor, why are these two pairs relatively dissimilar?

The mean pairwise identity provided in **Fig. S3B** is a measure of the quality of the reads and the quality of the mapping. Therefore, it does not reflect the similarity of the isolates.

7. Fig. 5. In panel A, this reviewer is not familiar with the terms "sense" and "nonsense" being used in this way. In panel B, different colors should be used for the different lines to make it easier to distinguish them.

The terms "sense" and "nonsense" have been replaced in the figure legend. Similarly, the colors for the different lines in panel B have been changed as requested.

Reviewer #2

Interesting work analyzing the evolution of *Pseudomonas aeruginosa* during a prolonged outbreak in burn ICU. There are however some comments for the authors' consideration:

We thank reviewer #2 for the interest on our work.

1. Lines 140-144. Why were only the MICs of imipenem and meropenem determined? Would be informative to add other relevant antipseudomonal agents.

We originally focused on the MICs of imipenem and meropenem due to the striking genetic signature at the *oprD* locus. In response to the reviewer's comment, we obtained the antibiotic susceptibility profiles for other relevant anti-*Pseudomonas* agents, including penicillins, cephalosporins, quinolones and colistin.

Data collected showed emergence of resistance also against other antibiotic classes. This information has been added to the Supplemental material (**Fig. S4**) and some commentary added to the text (248-251).

2. Related to this, other relevant mutations related to efflux pumps expression (*mexR*, *mexZ*, *mexS*) or AmpC expression (*dacB*, *mpl*, *ampR*) are seen and should be interpreted since many of the mutated genes are related to the *P. aeruginosa* mutational resistome (see for instance López-Causapé C front Microbiol 2018).

We agree with the reviewer. We added some text and also the reference (lines 248-251).

3. Lines 240-245. The interpretation of EUCAST breakpoints and definitions is incorrect, particularly "intermediately susceptible" shall not be used since it is misleading.

We thank the reviewer for pointing out this important correction. Isolates that we previously characterized as "intermediately susceptible" have been adapted according to the new definition of "I" (i.e., "susceptible increased exposure") throughout the text and in **Fig. 5**.

4. One relevant phenotype that could/should be studied in the context of adaptation in burn wound infections is the capacity to form biofilms.

This is certainly an interesting point – In response to the comment, we evaluated biofilm formation for each of the 47 unique *P. aeruginosa* isolates. Biofilm capacity was assessed using a microtiter plate assay with crystal violet staining (1). The assay was performed in technical duplicates and in biological triplicates. To determine the differences in biofilm capacity, we compared the biomass produced by the core isolate (ie., H26076) with the biomass produced by each unique isolate in the collection. Our data did not reveal a statistically significant difference in biofilm forming capacity for any of the isolates ($p \geq 0.9899$ for each comparison, using Kruskal Wallis with Dunn's multiple comparison test for non-normally distributed data). Indeed, the methodological approach has well known limitations. We feel that the biofilm formation capacity of the isolates may warrant further investigations using more elegant methods (2), however this falls outside the scope of the current report.

5. It might be informative to discuss your results together with those of Wheatley R Nat Commun 2021.

Some discussion regarding the results obtained by Wheatley R Nat Commun 2021 has been added in the text (lines 304-306).

In light of these improvements, we believe the manuscript is now ready for publication in Microbiology Spectrum.

Respectfully yours,

Prof. Yok Ai Que
Senior Physician

References

1. O'Toole GA. Microtiter dish biofilm formation assay. *Journal of visualized experiments : JoVE* 2011.
2. Wilson C, Lukowicz R, Merchant S, Valquier-Flynn H, Caballero J, Sandoval J, Okuom M, Huber C, Brooks TD, Wilson E, Clement B, Wentworth CD, Holmes AE. Quantitative and Qualitative Assessment Methods for Biofilm Growth: A Mini-review. *Res Rev J Eng Technol* 2017; 6.

October 19, 2022

Prof. Yok-Ai Que
Inselspital
Bern
Switzerland

Re: Spectrum02743-22R1 (Parallel evolution of *Pseudomonas aeruginosa* during a prolonged ICU infection outbreak)

Dear Prof. Yok-Ai Que:

I am sorry this has taken a while to get back to you, but I am delighted to inform you that your manuscript has finally been accepted to Microbiology Spectrum, and I am forwarding it to the ASM Journals Department for publication. You will be notified when your proofs are ready to be viewed.

Sincerely,

Joanna Goldberg
Editor, Microbiology Spectrum

Journals Department
Supplemental Material: Accept
Supplemental Material: Accept
Supplemental Material: Accept
Supplemental Material: Accept
Supplemental Material: Accept